# The state of the leprosy epidemic in Yunnan, China 2011–2020: A spatial and spatiotemporal analysis, highlighting areas for intervention

Xiaohua Chen[1,2], Tie-Jun Shui[3]*

**1** Beijing Tropical Medicine Research Institute, Beijing Friendship Hospital, Capital Medical University, Beijing, China, **2** Beijing Key Laboratory for Research on Prevention and Treatment of Tropical Diseases, Capital Medical University, Beijing, China, **3** Yunnan Center for Disease Control and Prevention, Yunnan, China

* 67637594@qq.com

**Data Availability Statement:** All relevant data are within the manuscript.

**Funding:** This study was funded by Health Commission of Yunnan Province (No:2017NS098)

## Abstract

### Background

Despite public health efforts to reduce the leprosy burden in Yunnan, China, leprosy remains an important public health problem in some specific areas. We analyzed the epidemiological characteristics and spatial distribution of leprosy in Yunnan, China, and provide data to guide disease prevention and control efforts.

### Methodology/principal findings

The surveillance data of newly detected leprosy cases in Yunnan, China, during 2011–2020 were extracted from the LEPROSY MANAGEMANT INFORMATION SYSTEM IN CHINA (LEPMIS), and spatial distribution analysis, spatial autocorrelation analysis, and spatiotemporal scanning were performed with ArcGIS 10.6.1, GeoDa 1.8.8, and SaTScan 9.4.3 software, respectively. A total of 1907 newly detected leprosy cases were reported in Yunnan, China, during 2011–2020. The new case detection rate (NCDR) decreased from 0.62 in 2011 to 0.25 in 2020, with an annual incidence of 0.41/100,000 population. The proportions of multibacillary (MB) cases, cases in female patients, cases causing grade 2 physical disability (G2D), and cases in pediatric patients were 67.07%, 33.93%, 17.99%, and 2.83%, respectively. The number of counties with an incidence above 1/100,000 population decreased from 30 in 2011 to 8 in 2020. The Moran's *I* of leprosy in Yunnan, China, during 2011–2020 ranged from 0.076 to 0.260, indicating the presence of spatial clusters. Local spatial autocorrelation (LSA) analysis showed that high-high cluster areas (hot spots) were mainly distributed in the southeastern, northern, and northwestern regions. Spatiotemporal scanning showed three clusters with high NCDRs. The probably primary clusters, occurring during January 1, 2011–December 31, 2015, covered 11 counties in the southeastern region (RR = 5.046515, LRR = 271.749664, P = 0.000).

by TS.The funder played no role in study design, data collection and analysis, decision to publish, and preparation of the manuscript.

**Competing interests:** The authors have declared that no competing interests exist.

## Conclusion

The number of leprosy cases in Yunnan decreased overall, although some high-NCDR regions remained. Geographic information system (GIS) analysis coupled with spatial analysis indicated regions with leprosy clusters. Continuous leprosy prevention and control strategies in Yunnan Province should be established, and interventions in high-risk regions should be prioritized and further strengthened.

### Author summary

China has achieved the goal of leprosy elimination established by the WHO. The overall incidence and prevalence rates of leprosy indicate a low endemic level in China. However, there are still specific areas with leprosy transmission in some parts of China, especially Yunnan Province. This study aimed to reveal the epidemic state and identify spatial and spatiotemporal clusters of leprosy in Yunnan, China, from 2011–2020. A total of 1907 newly detected cases were identified; 67.07% were MB and 32.93% were PB leprosy cases. Males were predominant (66.07%), 17.99% of patients presented with G2D, and 2.83% of patients were under 15 years old. Performed with ArcGIS 10.6.1, GeoDa 1.8.8, and SaTScan 9.4.3 software, three significant spatial clusters (hot spots) and three significant spatiotemporal clusters (high-risk areas) were observed. These results highlight the at-risk areas for prioritization and further intervention.

## Introduction

Leprosy is a chronic infectious disease caused by *Mycobacterium leprae*, which mainly affects the skin and peripheral nerves [1] and causes permanent disability and social stigma [2].

Despite being declared "eliminated" by the World Health Organization (WHO) in 2000, the global prevalence rate is <1 case/10,000 people, and leprosy remains an important public health problem in some low- and middle-income countries [3] and some particular high-burden areas [4,5], such as Indian Brazil and Indonesia; these regions took many more years to reach the national elimination target of <1 case per 10,000 people and accounted for most of the new cases (80.20%) during that time [6].

After 70 years of implementation of a multifaceted strategy, the overall incidence and prevalence rates of leprosy have steadily declined in China [7–10]. A similar trend was also reported in Yunnan Province [6,7]. Multidrug therapy (MDT) and multiple control strategies have been applied to eliminate leprosy in China over the past 30 years [11]. MDT, comprising rifampicin, clofazimine and dapsone, was recommended by the WHO in the 1980s [12] and has proven highly effective. MDT was introduced in China in 1983 [13] in Yunnan [14]; its application was then expanded to the whole province and whole country by the end of the 1980s. In 2004, a special fund for leprosy was established in the region by the central government, and from 2011 to 2020, a leprosy elimination program (2011–2020) was initiated in Yunnan and other provinces of China to promote eradication of the disease [11]. However, there were still small areas with endemic leprosy in some parts of China, especially southwestern China, which has relatively high leprosy endemicity and includes some parts of Yunnan Province [15,16].

In recent years, spatial and spatiotemporal analyses have been widely applied to describe the distribution characteristics and transmission patterns of leprosy in China [17] and other countries [4,18–28]. These studies demonstrated that spatial analysis could identify clusters of leprosy; this, spatial analysis seems to be a very useful tool to study leprosy and guide interventions and surveillance [2]. Few studies have been conducted in Southwest China to explore the spatial epidemiological characteristics at the county level. To improve leprosy control measures, we conducted geographical information system (GIS)-based spatiotemporal scan analysis in Yunnan from 2011 to 2020.

## Methods

### Ethics statement

This study was approved by the ethics committee of the Yunnan Center for Disease Control and Prevention, Yunnan, China. Individual identifying information was not available and therefore not included in the study.

### Study area

Yunnan has the highest burden of leprosy in China. Yunnan Province is located on the southwest boundary of China and is bordered by Myanmar to the west, Laos to the south, and Vietnam to the southeast as well as the Chinese provinces and regions of Guangxi Zhuang Autonomous Region and Guizhou Province to the east, Sichuan Province to the north, and Tibet Autonomous Region to the northwest. Yunnan is a mountainous province with 16 districts and 129 counties. It covers an area of 394,000 km$^2$, of which 94% is mountains, hills, and valleys, and only 6% is plains. Its population is 48.3 million according to the 2018 census.

### Subjects

The surveillance data of newly detected leprosy cases in Yunnan, China, from January 1, 2011, to December 31, 2020, were extracted from the LEPROSY MANAGEMANT INFORMATION SYSTEM (LEPMIS) database in China. The data of newly detected leprosy cases were obtained from the Yunnan Center for Disease Control and Prevention, and the data were permitted to be used by the Yunnan Center for Disease Control and Prevention. The diagnosis criteria were based on clinical, bacteriological, and histopathological profiles [29]. According to the WHO operational classification, newly detected leprosy cases were classified as multibacillary (MB) or paucibacillary (PB). The basic demographic data of the patients were extracted from the LEPMIS; the collected data included age, sex, province, prefecture, county, date of diagnosis, Ridley-Jopling classification, and WHO operational classification.

### Population data

The population data of the study area were obtained from the National Bureau of Statistics of the People's Republic of China.

### Statistical analysis

Excel 2007 was used to compile the data of newly detected leprosy cases; calculate the age of patients according to their birth date and the diagnosis date; and described the basic demographic characteristics, time distribution trends and regional distribution characteristics of cases. The data were subsequently analyzed using GraphPad Prism version 6 (GraphPad Software, La Jolla, California, USA). The new case detection rate (NCDR) was defined as the

number of newly detected cases per year per 100,000 general population. Grade 2 disability (G2D) was defined as visible disability [30].

### Spatial and spatiotemporal analyses

The geographical distribution of newly detected leprosy cases was mapped by ArcGIS software version 10.1 (Environmental Systems Research Institute, Inc., Redlands, CA, USA). The spatial autocorrelation analysis was performed in GeoDa 1.14.0.0 (Dr. Luc Anselin, Spatial Analysis Laboratory, Department of Agricultural and Consumer Economics, University of Illinois, Urbana Champaign). Spatial autocorrelation analysis is a spatial statistical method that can reveal the regional structure of spatial variables. It mainly includes global autocorrelation analysis and local autocorrelation analysis. Moran's $I$ values were used to identify global spatial autocorrelations and detect the spatial distribution patterns of newly detected leprosy cases in Yunnan, China. Local Getis $G_i^*$ statistics were calculated to identify local spatial autocorrelations and determine the locations of clusters or hot spots. SaTScan 9.5 (Kull-dorff, Boston, MA, USA), a spatial data processing software program, was used to identify the spatial patterns, temporal patterns and clusters of leprosy in different counties and during different time periods based on the Poisson probability model. A P-value less than 0.05 was considered significant.

## Results

### Basic information of new detected leprosy cases

A total of 1907 newly detected leprosy cases were reported in Yunnan Province during 2011–2020. The average NCDR was 0.41 per 100,000 population. The number of newly detected leprosy cases declined from 283 newly detected leprosy cases in 2011 to 119 cases in 2020. The NCDR declined from 0.62 per 100,000 population in 2011 to 0.25 per 100,000 population in 2020. Before 2014, the NCDRs were higher than 0.5 per 100,000 population, at 0.62, 0.50, and 0.52 per 100,000 population in 2011, 2012, and 2013, respectively. After 2014, the NCDRs were lower than 0.5 per 100,000 population, declining from 0.44 in 2014 to 0.25 per 100,000 population in 2020 (Table 1 and Fig 1).

**Table 1. Epidemiological Characteristics of Newly Detected Leprosy Cases in Yunnan, China, 2011–2020.**

| Year | New cases detected | NCDR | MB proportion | | Children proportion | | Female proportion | | G2D proportion | | Prevalence | | Relapsed cases detected |
|---|---|---|---|---|---|---|---|---|---|---|---|---|---|
| | n | per 100,000 population | n | % | n | % | n | % | n | % | n | per 100,000 population | n |
| 2011 | 283 | 0.62 | 172 | 60.78% | 7 | 2.47% | 103 | 36.40% | 83 | 29.33% | 1260 | 2.74 | 10 |
| 2012 | 230 | 0.50 | 147 | 63.91% | 10 | 4.35% | 72 | 31.30% | 54 | 23.48% | 1104 | 2.38 | 16 |
| 2013 | 241 | 0.52 | 155 | 64.32% | 6 | 2.49% | 81 | 33.61% | 43 | 17.84% | 1039 | 2.23 | 9 |
| 2014 | 208 | 0.44 | 136 | 65.38% | 6 | 2.88% | 68 | 32.69% | 38 | 18.27% | 912 | 1.95 | 10 |
| 2015 | 187 | 0.40 | 122 | 65.24% | 10 | 5.35% | 56 | 29.95% | 30 | 16.04% | 739 | 1.57 | 6 |
| 2016 | 170 | 0.36 | 109 | 64.12% | 8 | 4.71% | 56 | 32.94% | 28 | 16.47% | 608 | 1.28 | 11 |
| 2017 | 159 | 0.33 | 112 | 70.44% | 1 | 0.63% | 52 | 32.70% | 20 | 12.58% | 538 | 1.13 | 11 |
| 2018 | 174 | 0.36 | 129 | 74.14% | 3 | 1.72% | 67 | 38.51% | 18 | 10.34% | 477 | 0.99 | 9 |
| 2019 | 136 | 0.28 | 101 | 74.26% | 2 | 1.47% | 51 | 37.50% | 17 | 12.50% | 377 | 0.78 | 6 |
| 2020 | 119 | 0.25 | 96 | 80.67% | 1 | 0.84% | 41 | 34.45% | 12 | 10.08% | 306 | 0.64 | 9 |
| Total | 1907 | 0.41 | 1279 | 67.07% | 54 | 2.83% | 647 | 33.93% | 343 | 17.99% | / | 1.57 | 97 |

NCDR: new cases detected rate per 100,000 population; MB: multibacillary; G2D: grade 2 disability.

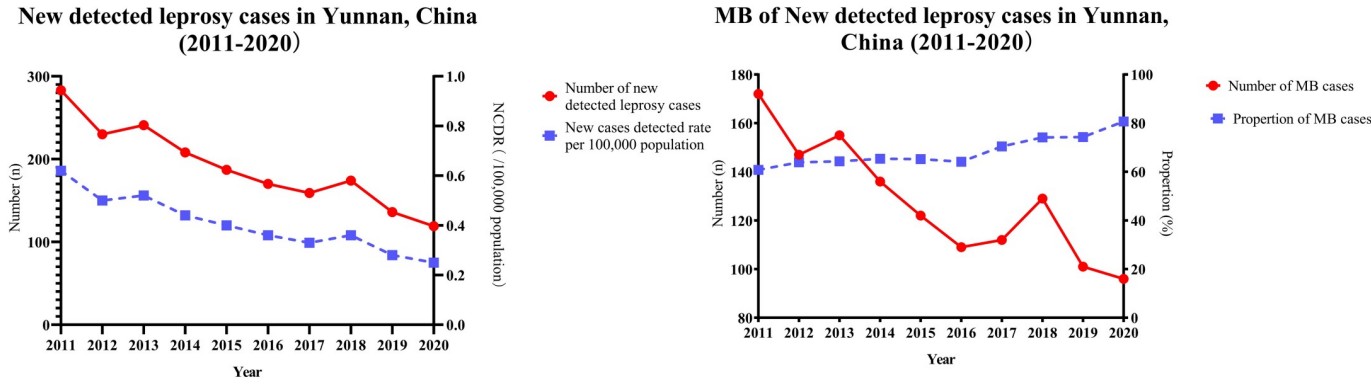

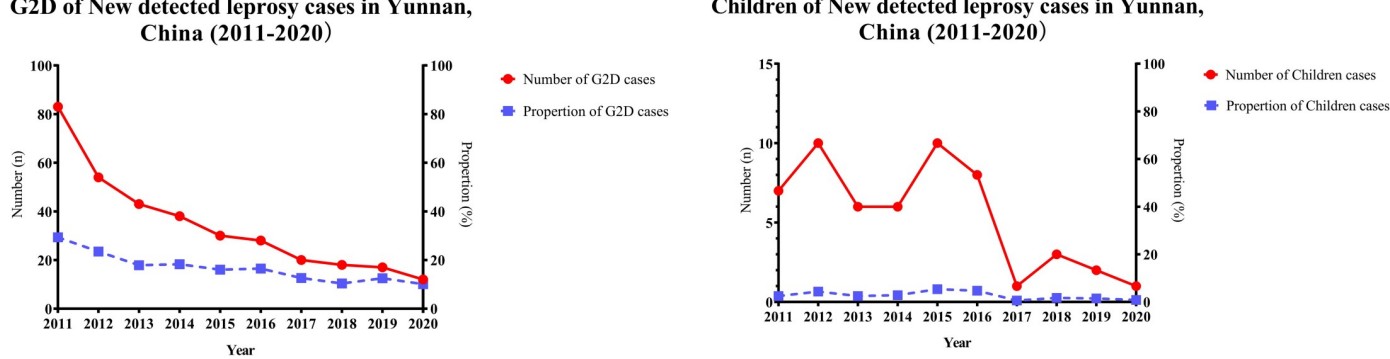

**Fig 1. Epidemiological Characteristics of new detected leprosy cases in Yunnan, China, 2011–2020.**

A total of 1279 MB cases were reported in Yunnan Province during 2011–2020, accounting for 67.07% of newly detected leprosy cases in the whole province. The number of MB cases declined annually from 172 in 2011 to 96 in 2020, while the ratio of MB cases increased annually from 60.78% in 2011 to 80.67% in 2020 (Table 1 and Fig 1).

From 2011 to 2020, a total of 343 patients with newly diagnosed leprosy in Yunnan Province had G2D. The total number of patients with G2D among patients with newly detected cases decreased from 83 in 2011 to 12 in 2020. During the same period, the rate of G2D showed a decreasing trend, but there were small fluctuations. In 2011, the rate of G2D was the highest (29.33%). The rate of G2D gradually decreased to 10.34% in 2018, increased to 12.50% in 2019, and further decreased to 10.08% in 2020 (Table 1 and Fig 1).

From 2011 to 2020, 54 cases of leprosy in children were reported in Yunnan Province, accounting for 2.8% of the newly detected leprosy cases in the province. There were three peaks in the number of children with leprosy in Yunnan Province in 2012, 2015 and 2016, respectively, and the numbers of new cases registered during the period were 10, 10 and 8, accounting for 4.35%, 5.35% and 4.71% of the total number of new cases, respectively. The prevalence of leprosy among children fluctuated from 0.63% to 5.35%. As of 2020, leprosy in children (n = 1) was still being reported in Yunnan Province (Table 1 and Fig 1).

From 2011 to 2020, 647 cases of leprosy in females were reported in Yunnan Province, accounting for 33.93% of the new cases in the province, and the ratio of males to females was 1.95:1. The number of new cases of leprosy among females decreased from 103 in 2011 to 41

in 2020, with the proportion of female leprosy cases fluctuating between 29.95% and 38.51% (Table 1 and Fig 1).

The number of active cases in Yunnan Province decreased annually from 1260 cases in 2011 to 306 cases in 2020. The prevalence rate also decreased annually during the same period, from 2.74/100,000 population in 2011 to 0.64/100,000 population in 2020 (Table 1). From 2011 to 2020, 97 recurrent cases of leprosy were reported in Yunnan Province (Table 1).

## Spatial distribution of newly detected leprosy cases by ArgGIS

The spatial distribution of newly detected leprosy cases is shown in Table 2 and Fig 2. At the province level, the NCDR in Yunnan steadily declined from 0.62 per 100,000 population in 2011 to 0.25 per 100,000 population in 2020, as described previously. At the prefecture level, the NCDR in Wenshan Zhuang and Miao Autonomous Prefecture decreased from 1.86 in 2011 to 0.77 per 100,000 population in 2017 and rebounded to 0.96, 0.99, and 0.94 per 100,000 population in 2018, 2019, and 2020, respectively. The 10-year average NCDR was 1.20 per 100,000 population, and this prefecture consistently had the highest rate in the province (Table 2). In three districts, the NCDRs rebounded to more than 1 per 100,000 population, reaching 1.56 in 2015 in Xishuangbanna Dai Autonomous Prefecture, 1.68 in 2017 in Dêqên Tibetan Autonomous Prefecture, and 1.14 per 100,000 population in 2018 Pu'er city. The NCDRs in Xishuangbanna Dai Autonomous Prefecture and Pu'er city decreased rapidly to 0.51 in 2020 and 0.46 per 100,000 population in 2020, and the NCDR in Dêqên Tibetan Autonomous Prefecture was 0.94 per 100,000 population in 2020 after three years of slow decline (Table 2). The average NCDRs in Wenshan Zhuang and Miao Autonomous Prefecture and Dêqên Tibetan Autonomous Prefecture were 1.20 and 1.01 per 100,000 population during the past 10 years, respectively. At the county level, 30 out of 129 counties had an NCDR above 1 per 100,000 population in 2011. These numbers decreased in 8 counties in the province in 2020, and the highest NCDRs were found in Dêqên County, Yanshan, Qiubei, Kaiyuan, Yuanmou County, and Maguan as well as Jinggu Dai and Yi Autonomous County and Lancang Lahu Autonomous County in 2020 (Table 2 and Fig 2).

## Spatial autocorrelation analysis by GeoDa

The global spatial autocorrelation analysis showed that the annual Moran's *I* values of the NCDRs from 2011 to 2020 were significantly different, except in 2016 (Table 3), indicating that the NCDR of leprosy in Yunnan was nonrandomly distributed, and the distribution of leprosy in Yunnan was spatially autocorrelated over the 10-year study period. The Moran's *I* values of the annual NCDRs of leprosy were positive, and the P values were less than 0.05, except in 2016. There was a positive global spatial autocorrelation among the NCDRs of leprosy in most years.

The local spatial autocorrelation analysis showed different clusters according to the LISA analysis (Fig 3). From 2011 to 2020, the NCDRs of new leprosy cases in Yunnan Province revealed three types of clusters: high-high areas (hot spots), low-low areas (cold spots) and low-high areas. The high-high areas (hot spots) were mainly concentrated in the southeastern and north-northwest areas of Yunnan Province, while the low-low areas (cold spots) were concentrated in the central, northeastern, western and southwestern areas (Table 4 and Fig 3).

## Spatiotemporal clustering analysis by SaTScan

The NCDRs of leprosy during 2011–2020 were analyzed with spatiotemporal scanning using SaTScan. The results showed that the NCDRs of leprosy were spatiotemporally clustered. One probably primary cluster, 1 secondary cluster, 1 tertiary cluster and 1 quaternary cluster

**Table 2. Spatial distribution of newly detected leprosy cases in Yunnan, China, 2011–2020.**

| Region | NCDR (y) | | | | | | | | | | |
|---|---|---|---|---|---|---|---|---|---|---|---|
| | 2011 | 2012 | 2013 | 2014 | 2015 | 2016 | 2017 | 2018 | 2019 | 2020 | 2011–2020 |
| **Yunnan province** | **0.62** | **0.50** | **0.52** | **0.44** | **0.40** | **0.36** | **0.33** | **0.36** | **0.28** | **0.25** | **0.41** |
| **Kunming City** | **0.37** | **0.29** | **0.31** | **0.31** | **0.18** | **0.25** | **0.18** | **0.18** | **0.06** | **0.06** | **0.22** |
| Wuhua District | 0.23 | 0.00 | 0.12 | 0.23 | 0.12 | 0.00 | 0.00 | 0.00 | 0.11 | 0.00 | 0.08 |
| Panlong District | 0.12 | 0.12 | 0.00 | 0.25 | 0.00 | 0.12 | 0.00 | 0.00 | 0.00 | 0.00 | 0.06 |
| Guandu District | 0.00 | 0.24 | 0.12 | 0.35 | 0.12 | 0.34 | 0.00 | 0.00 | 0.11 | 0.11 | 0.14 |
| Xishan District | 0.26 | 0.53 | 0.40 | 0.40 | 0.00 | 0.13 | 0.00 | 0.25 | 0.00 | 0.00 | 0.20 |
| Dongchuan District | 0.00 | 0.00 | 0.36 | 0.00 | 0.00 | 0.36 | 0.35 | 0.00 | 0.00 | 0.00 | 0.11 |
| Chenggong District | 0.32 | 0.32 | 0.00 | 0.31 | 0.31 | 0.00 | 0.00 | 0.00 | 0.00 | 0.00 | 0.13 |
| Jinning District | 0.35 | 0.00 | 0.69 | 0.00 | 0.00 | 0.00 | 0.33 | 0.65 | 0.32 | 0.00 | 0.23 |
| Fumin County | 1.36 | 0.68 | 2.69 | 1.34 | 0.67 | 0.66 | 0.00 | 1.28 | 0.00 | 0.64 | 0.93 |
| Yiliang County | 0.24 | 0.47 | 0.23 | 0.23 | 0.00 | 0.23 | 0.90 | 0.00 | 0.00 | 0.00 | 0.23 |
| Shilin Yi Autonomous County | 0.00 | 0.40 | 0.00 | 0.00 | 0.00 | 0.00 | 0.38 | 0.38 | 0.00 | 0.00 | 0.12 |
| Songming County | 0.35 | 0.34 | 0.00 | 0.00 | 0.33 | 0.33 | 0.33 | 0.00 | 0.00 | 0.00 | 0.17 |
| Luquan Yi and Miao Autonomous County | 1.25 | 0.75 | 0.49 | 0.49 | 0.49 | 1.19 | 0.71 | 0.00 | 0.00 | 0.00 | 0.54 |
| Xundian Hui and Yi Autonomous County | 1.52 | 0.65 | 0.64 | 0.64 | 0.84 | 0.41 | 0.21 | 0.64 | 0.21 | 0.21 | 0.60 |
| Anning City | 0.29 | 0.00 | 0.58 | 0.29 | 0.29 | 0.28 | 0.00 | 0.53 | 0.00 | 0.26 | 0.25 |
| **Qujing City** | **0.15** | **0.22** | **0.13** | **0.07** | **0.08** | **0.05** | **0.15** | **0.13** | **0.10** | **0.07** | **0.12** |
| Kirin district | 0.27 | 0.13 | 0.00 | 0.13 | 0.00 | 0.00 | 0.00 | 0.26 | 0.00 | 0.00 | 0.08 |
| Zhanyi District | 0.23 | 0.23 | 0.46 | 0.23 | 0.23 | 0.00 | 0.00 | 0.00 | 0.11 | 0.00 | 0.15 |
| Malone District | 0.54 | 0.00 | 0.53 | 0.00 | 0.00 | 0.00 | 1.04 | 0.00 | 0.00 | 0.00 | 0.21 |
| Luliang County | 0.00 | 0.16 | 0.16 | 0.00 | 0.00 | 0.00 | 0.00 | 0.00 | 0.00 | 0.00 | 0.03 |
| Shizong County | 0.00 | 0.51 | 1.00 | 0.50 | 0.74 | 0.25 | 0.73 | 0.24 | 0.31 | 0.24 | 0.45 |
| Luoping County | 0.18 | 0.36 | 0.00 | 0.00 | 0.18 | 0.18 | 0.35 | 0.00 | 0.49 | 0.17 | 0.19 |
| Fuyuan County | 0.00 | 0.00 | 0.00 | 0.00 | 0.00 | 0.00 | 0.13 | 0.00 | 0.00 | 0.00 | 0.01 |
| Huize County | 0.22 | 0.55 | 0.00 | 0.00 | 0.00 | 0.00 | 0.00 | 0.43 | 0.13 | 0.21 | 0.15 |
| Xuanwei City | 0.15 | 0.08 | 0.00 | 0.00 | 0.00 | 0.07 | 0.07 | 0.07 | 0.00 | 0.00 | 0.04 |
| **Yuxi City** | **0.17** | **0.13** | **0.17** | **0.13** | **0.00** | **0.08** | **0.29** | **0.17** | **0.04** | **0.04** | **0.12** |
| Hongta District | 0.00 | 0.00 | 0.20 | 0.00 | 0.00 | 0.00 | 0.20 | 0.00 | 0.00 | 0.00 | 0.04 |
| Jiangchuan District | 0.00 | 0.00 | 0.00 | 0.00 | 0.00 | 0.00 | 0.00 | 0.00 | 0.00 | 0.00 | 0.00 |
| Chengjiang County | 0.00 | 0.00 | 0.59 | 0.00 | 0.00 | 0.00 | 0.00 | 0.55 | 0.00 | 0.00 | 0.11 |
| Tonghai County | 0.66 | 0.00 | 0.00 | 0.00 | 0.00 | 0.00 | 0.00 | 0.32 | 0.00 | 0.00 | 0.10 |
| Huaning County | 0.46 | 1.40 | 0.93 | 0.47 | 0.00 | 0.00 | 1.36 | 0.90 | 0.00 | 0.00 | 0.55 |
| Yimen County | 0.56 | 0.00 | 0.00 | 0.00 | 0.00 | 0.00 | 1.10 | 0.00 | 0.00 | 0.55 | 0.22 |
| Eshan Yi Autonomous County | 0.00 | 0.00 | 0.00 | 0.00 | 0.00 | 0.00 | 0.00 | 0.00 | 0.00 | 0.00 | 0.00 |
| Xinping Yi and Dai Autonomous County | 0.00 | 0.00 | 0.00 | 0.35 | 0.00 | 0.00 | 0.00 | 0.00 | 0.00 | 0.00 | 0.04 |
| Yuanjiang Hani, Yi and Dai Autonomous County | 0.00 | 0.00 | 0.00 | 0.46 | 0.00 | 0.90 | 0.45 | 0.00 | 0.44 | 0.00 | 0.23 |
| **Baoshan City** | **0.36** | **0.24** | **0.32** | **0.31** | **0.27** | **0.43** | **0.08** | **0.23** | **0.19** | **0.00** | **0.24** |
| Longyang District | 0.53 | 0.21 | 0.53 | 0.63 | 0.52 | 0.72 | 0.00 | 0.41 | 0.20 | 0.00 | 0.38 |
| Shidian County | 0.65 | 0.65 | 0.00 | 0.00 | 0.64 | 0.00 | 0.00 | 0.31 | 0.31 | 0.00 | 0.26 |
| Longling County | 0.00 | 0.00 | 0.00 | 0.00 | 0.00 | 0.00 | 0.00 | 0.00 | 0.28 | 0.00 | 0.03 |
| Changning County | 0.29 | 0.29 | 0.57 | 0.28 | 0.00 | 0.56 | 0.00 | 0.28 | 0.00 | 0.00 | 0.23 |
| Tengchong City | 0.15 | 0.15 | 0.15 | 0.15 | 0.00 | 0.30 | 0.30 | 0.00 | 0.34 | 0.00 | 0.15 |
| **Zhaotong City** | **0.53** | **0.34** | **0.28** | **0.34** | **0.17** | **0.22** | **0.24** | **0.20** | **0.18** | **0.20** | **0.27** |
| Zhaoyang District | 0.63 | 0.13 | 0.13 | 0.12 | 0.00 | 0.12 | 0.12 | 0.00 | 0.12 | 0.12 | 0.15 |
| Ludian County | 0.25 | 1.27 | 0.25 | 1.49 | 0.25 | 0.49 | 0.48 | 0.00 | 0.47 | 0.48 | 0.54 |
| Qiaojia County | 0.96 | 0.00 | 0.00 | 0.38 | 0.38 | 0.94 | 0.75 | 1.11 | 0.37 | 0.37 | 0.53 |

*(Continued)*

**Table 2.** (Continued)

| Region | NCDR (y) | | | | | | | | | | |
|---|---|---|---|---|---|---|---|---|---|---|---|
| | 2011 | 2012 | 2013 | 2014 | 2015 | 2016 | 2017 | 2018 | 2019 | 2020 | 2011–2020 |
| Yanjin County | 0.00 | 0.00 | 0.00 | 0.00 | 0.00 | 0.00 | 0.00 | 0.00 | 0.00 | 0.00 | 0.00 |
| Daguan County | 1.51 | 1.89 | 1.51 | 0.37 | 0.00 | 0.00 | 1.09 | 0.00 | 0.00 | 0.00 | 0.64 |
| Yongshan County | 1.51 | 0.76 | 0.00 | 0.75 | 0.75 | 0.49 | 0.00 | 0.48 | 0.47 | 0.72 | 0.59 |
| Suijiang County | 0.00 | 0.00 | 0.00 | 0.65 | 0.64 | 0.00 | 0.00 | 0.62 | 0.00 | 0.00 | 0.19 |
| Zhenxiong County | 0.15 | 0.30 | 0.37 | 0.15 | 0.14 | 0.07 | 0.07 | 0.07 | 0.14 | 0.21 | 0.17 |
| Yiliang County | 0.95 | 0.00 | 0.76 | 0.19 | 0.00 | 0.19 | 0.37 | 0.18 | 0.18 | 0.00 | 0.28 |
| Weixin County | 0.00 | 0.00 | 0.00 | 0.25 | 0.00 | 0.00 | 0.00 | 0.00 | 0.00 | 0.00 | 0.03 |
| Shuifu City | 0.00 | 0.00 | 0.00 | 0.00 | 0.00 | 0.00 | 0.00 | 0.00 | 0.00 | 0.00 | 0.00 |
| **Lijiang City** | **0.64** | **0.64** | **0.95** | **0.31** | **0.63** | **0.39** | **0.47** | **0.08** | **0.46** | **0.54** | **0.51** |
| Gucheng District | 0.00 | 0.00 | 0.00 | 0.00 | 0.93 | 0.00 | 0.46 | 0.00 | 0.00 | 0.91 | 0.23 |
| Yulong Naxi Autonomous County | 0.46 | 0.00 | 0.00 | 0.00 | 0.00 | 0.00 | 0.00 | 0.00 | 0.00 | 0.00 | 0.05 |
| Yongsheng County | 0.76 | 1.01 | 1.77 | 1.00 | 0.50 | 0.25 | 0.25 | 0.25 | 0.25 | 0.49 | 0.65 |
| Huaping County | 1.18 | 1.77 | 1.77 | 0.00 | 1.74 | 1.74 | 1.74 | 0.00 | 2.28 | 0.57 | 1.28 |
| Ninglang Yi Autonomous County | 0.77 | 0.38 | 0.75 | 0.00 | 0.37 | 0.37 | 0.37 | 0.00 | 0.37 | 0.74 | 0.41 |
| **Pu'er City** | **0.70** | **0.78** | **0.89** | **0.42** | **0.50** | **0.46** | **0.31** | **1.14** | **0.42** | **0.46** | **0.61** |
| Simao District | 0.00 | 0.34 | 0.00 | 0.00 | 0.67 | 0.00 | 0.33 | 0.00 | 0.32 | 0.32 | 0.20 |
| Ning'er Hani and Yi Autonomous County | 1.60 | 0.53 | 0.00 | 1.06 | 0.00 | 0.00 | 0.00 | 0.00 | 0.00 | 0.52 | 0.37 |
| Mojiang Hani Autonomous County | 0.00 | 0.55 | 1.08 | 0.27 | 0.54 | 0.00 | 0.27 | 0.27 | 0.00 | 0.54 | 0.35 |
| Jingdong Yi Autonomous County | 0.83 | 1.10 | 0.55 | 0.00 | 0.27 | 0.27 | 0.54 | 1.08 | 0.54 | 0.00 | 0.52 |
| Jinggu Dai and Yi Autonomous County | 0.34 | 0.00 | 0.00 | 0.67 | 0.66 | 0.67 | 0.33 | 0.00 | 0.00 | 1.00 | 0.37 |
| Zhenyuan Yi, Hani and Lahu Autonomous County | 0.48 | 0.47 | 0.94 | 0.47 | 0.00 | 0.00 | 0.00 | 0.47 | 0.00 | 0.00 | 0.28 |
| Jiangcheng Hani and Yi Autonomous County | 0.81 | 0.00 | 0.00 | 0.80 | 0.00 | 0.00 | 0.00 | 0.00 | 0.00 | 0.00 | 0.16 |
| Menglian Dai, Lahu and Va Autonomous County | 0.00 | 0.00 | 2.92 | 0.00 | 1.44 | 0.00 | 0.00 | 0.70 | 0.70 | 0.00 | 0.58 |
| Lancang Lahu Autonomous County | 1.41 | 2.01 | 2.19 | 0.79 | 0.79 | 1.78 | 0.20 | 3.79 | 1.19 | 1.00 | 1.52 |
| Ximeng Va Autonomous County | 2.18 | 1.08 | 0.00 | 0.00 | 0.00 | 0.00 | 2.12 | 4.18 | 1.04 | 0.00 | 1.06 |
| **Lincang City** | **0.49** | **0.41** | **0.28** | **0.32** | **0.40** | **0.32** | **0.12** | **0.08** | **0.04** | **0.08** | **0.25** |
| Linxiang District, | 0.92 | 0.92 | 0.61 | 0.92 | 0.61 | 0.91 | 0.30 | 0.00 | 0.00 | 0.30 | 0.55 |
| Fengqing County | 1.30 | 0.65 | 0.43 | 0.43 | 0.43 | 0.21 | 0.21 | 0.21 | 0.00 | 0.21 | 0.41 |
| Yun County | 0.00 | 0.44 | 0.22 | 0.44 | 0.65 | 0.22 | 0.21 | 0.21 | 0.00 | 0.00 | 0.24 |
| Yongde County | 0.27 | 0.27 | 0.00 | 0.27 | 0.00 | 0.53 | 0.00 | 0.00 | 0.00 | 0.00 | 0.13 |
| Zhenkang County | 0.00 | 0.00 | 0.00 | 0.00 | 0.00 | 0.00 | 0.00 | 0.00 | 0.00 | 0.00 | 0.00 |
| Shuangjiang Lahu, Va, Blang and Dai Autonomous County | 0.56 | 0.00 | 0.00 | 0.00 | 0.00 | 0.00 | 0.00 | 0.00 | 0.00 | 0.00 | 0.06 |
| Gengma Dai and Va Autonomous County | 0.33 | 0.00 | 0.33 | 0.00 | 0.66 | 0.33 | 0.00 | 0.00 | 0.32 | 0.00 | 0.20 |
| Cangyuan Va Autonomous County | 0.00 | 0.56 | 0.55 | 0.00 | 0.55 | 0.00 | 0.00 | 0.00 | 0.00 | 0.00 | 0.17 |
| **Chuxiong Yi Autonomous Prefecture** | **0.63** | **0.77** | **0.55** | **0.58** | **0.40** | **0.29** | **0.44** | **0.40** | **0.36** | **0.22** | **0.46** |
| Chuxiong City | 0.34 | 1.01 | 0.33 | 0.50 | 0.16 | 0.33 | 0.00 | 0.17 | 0.17 | 0.00 | 0.30 |
| Shuangbai County | 0.62 | 0.00 | 1.23 | 0.00 | 0.60 | 0.61 | 0.00 | 0.00 | 0.00 | 0.00 | 0.31 |
| Mouding County | 0.00 | 0.47 | 0.47 | 0.46 | 0.46 | 0.00 | 0.93 | 0.47 | 0.47 | 0.00 | 0.37 |
| Nanhua County | 0.00 | 0.42 | 0.00 | 0.00 | 0.00 | 0.00 | 0.00 | 0.00 | 0.00 | 0.00 | 0.04 |
| Yao'an County | 0.50 | 0.00 | 0.00 | 0.49 | 0.00 | 0.49 | 0.98 | 0.49 | 0.49 | 0.00 | 0.34 |
| Dayao County | 0.36 | 1.09 | 1.08 | 0.72 | 1.42 | 0.36 | 0.72 | 1.07 | 0.71 | 0.72 | 0.83 |
| Yongren County | 1.82 | 2.72 | 0.00 | 0.90 | 0.00 | 0.90 | 0.00 | 0.90 | 0.90 | 0.00 | 0.81 |
| Yuanmou County | 2.78 | 0.46 | 0.91 | 1.81 | 1.35 | 0.46 | 1.36 | 1.81 | 0.90 | 1.36 | 1.32 |
| Wuding County | 1.09 | 0.73 | 1.09 | 1.08 | 0.36 | 0.00 | 0.36 | 0.00 | 0.36 | 0.36 | 0.54 |
| Lufeng County | 0.23 | 0.94 | 0.47 | 0.23 | 0.00 | 0.23 | 0.46 | 0.00 | 0.23 | 0.00 | 0.28 |
| **Honghe Hani and Yi Autonomous Prefecture** | **0.84** | **0.90** | **0.85** | **0.70** | **0.71** | **0.47** | **0.64** | **0.51** | **0.51** | **0.34** | **0.65** |

*(Continued)*

**Table 2.** (Continued)

| Region | NCDR (y) | | | | | | | | | | |
|---|---|---|---|---|---|---|---|---|---|---|---|
| | 2011 | 2012 | 2013 | 2014 | 2015 | 2016 | 2017 | 2018 | 2019 | 2020 | 2011–2020 |
| Gejiu City | 0.86 | 1.09 | 0.65 | 0.22 | 0.21 | 0.21 | 0.00 | 0.00 | 0.00 | 0.21 | 0.35 |
| Kaiyuan City | 3.08 | 4.00 | 2.15 | 2.44 | 3.33 | 1.20 | 1.80 | 1.19 | 1.48 | 1.49 | 2.22 |
| Mengzi City | 1.19 | 1.19 | 1.19 | 1.89 | 1.88 | 1.16 | 0.70 | 0.88 | 1.75 | 0.66 | 1.25 |
| Miller City | 0.18 | 0.37 | 1.09 | 1.63 | 1.26 | 2.26 | 3.54 | 3.19 | 0.95 | 0.53 | 1.50 |
| Pingbian Miao Autonomous County | 1.94 | 1.89 | 3.66 | 0.00 | 1.21 | 0.36 | 0.36 | 0.71 | 0.53 | 0.65 | 1.13 |
| Jianshui County | 0.56 | 0.19 | 0.37 | 0.37 | 0.00 | 0.00 | 1.19 | 0.00 | 0.64 | 0.00 | 0.33 |
| Shiping County | 0.00 | 0.00 | 0.00 | 0.00 | 0.00 | 0.00 | 0.54 | 0.00 | 0.00 | 0.32 | 0.09 |
| Luxi County | 0.74 | 0.99 | 0.49 | 0.49 | 0.49 | 0.72 | 0.24 | 0.24 | 0.47 | 0.00 | 0.49 |
| Yuanyang County | 2.00 | 1.00 | 0.75 | 0.50 | 0.49 | 0.00 | 0.24 | 0.24 | 0.24 | 0.24 | 0.57 |
| Red River County | 0.00 | 0.00 | 0.00 | 0.00 | 0.00 | 0.00 | 0.00 | 0.00 | 0.00 | 0.00 | 0.00 |
| Jinping Miao, Yao, and Dai Autonomous County | 0.00 | 0.84 | 0.55 | 0.00 | 0.00 | 0.00 | 0.00 | 0.00 | 0.26 | 0.00 | 0.17 |
| Lüchun County | 0.00 | 0.45 | 0.44 | 0.00 | 0.00 | 0.00 | 0.43 | 0.00 | 0.00 | 0.00 | 0.13 |
| Hekou Yao Autonomous County | 0.95 | 0.00 | 1.89 | 0.00 | 0.00 | 0.00 | 0.00 | 0.00 | 0.00 | 0.92 | 0.38 |
| **Wenshan Zhuang and Miao Autonomous Prefecture** | **1.86** | **1.18** | **1.60** | **1.53** | **1.19** | **1.00** | **0.77** | **0.96** | **0.99** | **0.94** | **1.20** |
| Wenshan City | 1.85 | 1.85 | 2.04 | 2.23 | 1.21 | 0.61 | 0.60 | 1.40 | 0.60 | 0.80 | 1.32 |
| Yanshan County | 3.43 | 0.64 | 1.27 | 1.89 | 1.26 | 1.05 | 1.05 | 0.42 | 1.25 | 2.51 | 1.48 |
| Xichou County | 1.17 | 0.78 | 0.39 | 0.77 | 0.38 | 0.38 | 1.15 | 1.14 | 0.38 | 0.38 | 0.69 |
| Malipo County | 0.72 | 0.36 | 0.36 | 1.77 | 0.70 | 0.00 | 0.35 | 0.35 | 0.00 | 0.35 | 0.50 |
| Maguan County | 1.62 | 1.08 | 1.33 | 0.79 | 0.79 | 1.32 | 1.58 | 0.26 | 1.31 | 1.05 | 1.11 |
| Qiubei County | 3.33 | 2.08 | 4.54 | 3.28 | 2.85 | 2.25 | 1.02 | 2.03 | 3.02 | 1.62 | 2.60 |
| Guangnan County | 1.14 | 1.13 | 1.00 | 0.62 | 1.11 | 1.12 | 0.37 | 1.23 | 0.74 | 0.49 | 0.90 |
| Funing County | 1.22 | 0.97 | 0.97 | 0.96 | 0.48 | 0.48 | 0.48 | 0.24 | 0.00 | 0.00 | 0.58 |
| **Xishuangbanna Dai Autonomous Prefecture** | **1.23** | **0.79** | **0.79** | **0.52** | **1.56** | **1.20** | **0.77** | **0.85** | **0.51** | **0.51** | **0.87** |
| Jinghong City | 0.76 | 0.19 | 0.38 | 0.19 | 0.57 | 0.19 | 0.00 | 0.19 | 0.00 | 0.56 | 0.30 |
| Menghai County | 1.79 | 1.49 | 1.18 | 0.88 | 4.09 | 2.90 | 2.31 | 2.03 | 1.15 | 0.58 | 1.84 |
| Mengla County | 1.41 | 1.06 | 1.06 | 0.70 | 0.35 | 1.04 | 0.34 | 0.68 | 0.68 | 0.34 | 0.77 |
| **Dali Bai Autonomous Prefecture** | **0.72** | **0.40** | **0.51** | **0.45** | **0.14** | **0.34** | **0.25** | **0.25** | **0.08** | **0.20** | **0.33** |
| Dali City | 0.30 | 0.15 | 0.76 | 0.30 | 0.00 | 0.00 | 0.30 | 0.15 | 0.00 | 0.15 | 0.21 |
| Yangbi Yi Autonomous County | 0.00 | 0.00 | 0.00 | 0.94 | 0.00 | 0.94 | 0.00 | 0.00 | 0.93 | 0.93 | 0.37 |
| Xiangyun County | 2.62 | 0.43 | 0.63 | 0.21 | 0.00 | 1.05 | 0.00 | 0.42 | 0.00 | 0.21 | 0.56 |
| Binchuan County | 0.85 | 0.28 | 0.57 | 0.28 | 0.00 | 0.00 | 0.55 | 0.55 | 0.00 | 0.28 | 0.34 |
| Midu County | 0.63 | 0.00 | 0.00 | 0.31 | 0.00 | 0.00 | 0.00 | 0.00 | 0.00 | 0.00 | 0.09 |
| Nanjian Yi Autonomous County | 0.00 | 0.47 | 0.00 | 0.46 | 0.46 | 0.00 | 0.00 | 0.00 | 0.45 | 0.00 | 0.18 |
| Weishan Yi and Hui Autonomous County | 0.65 | 1.30 | 1.28 | 0.95 | 0.32 | 0.32 | 0.63 | 0.64 | 0.32 | 0.00 | 0.64 |
| Yongping County | 0.00 | 1.13 | 0.00 | 2.21 | 0.55 | 0.00 | 0.00 | 0.55 | 0.00 | 0.00 | 0.44 |
| Yunlong County | 0.50 | 0.49 | 0.98 | 0.00 | 0.48 | 0.97 | 0.00 | 0.00 | 0.00 | 0.48 | 0.39 |
| Eryuan County | 1.11 | 0.37 | 0.00 | 0.73 | 0.00 | 0.73 | 1.08 | 0.00 | 0.00 | 0.00 | 0.40 |
| Jianchuan County | 0.00 | 0.00 | 0.58 | 0.00 | 0.00 | 0.00 | 0.00 | 0.00 | 0.00 | 0.00 | 0.06 |
| Heqing County | 0.00 | 0.39 | 0.38 | 0.00 | 0.38 | 0.38 | 0.00 | 0.38 | 0.00 | 0.75 | 0.27 |
| **Dehong Dai and Jingpo Autonomous Prefecture** | **0.25** | **0.16** | **0.33** | **0.41** | **0.73** | **0.31** | **0.32** | **0.38** | **0.53** | **0.38** | **0.38** |
| Ruili City | 0.00 | 0.00 | 0.00 | 0.00 | 0.00 | 0.53 | 1.09 | 0.96 | 0.00 | 0.96 | 0.35 |
| Mang City | 0.51 | 0.51 | 1.01 | 1.01 | 1.50 | 0.73 | 0.50 | 0.48 | 1.18 | 0.48 | 0.79 |
| Lianghe County | 0.65 | 0.00 | 0.00 | 0.00 | 1.27 | 0.00 | 0.00 | 0.00 | 1.24 | 0.00 | 0.32 |
| Yingjiang County | 0.00 | 0.00 | 0.00 | 0.00 | 0.32 | 0.00 | 0.00 | 0.00 | 0.00 | 0.00 | 0.03 |
| Longchuan County | 0.00 | 0.00 | 0.00 | 0.54 | 0.00 | 0.00 | 0.00 | 0.51 | 0.00 | 0.51 | 0.16 |
| **Nujiang Lisu Autonomous Prefecture** | **0.00** | **0.19** | **0.18** | **0.00** | **0.00** | **0.00** | **0.00** | **0.18** | **0.18** | **0.00** | **0.07** |

*(Continued)*

**Table 2.** (Continued)

| Region | NCDR (y) | | | | | | | | | | |
|---|---|---|---|---|---|---|---|---|---|---|---|
| | 2011 | 2012 | 2013 | 2014 | 2015 | 2016 | 2017 | 2018 | 2019 | 2020 | 2011–2020 |
| Lushui City | 0.00 | 0.00 | 0.00 | 0.00 | 0.00 | 0.00 | 0.00 | 0.00 | 0.00 | 0.00 | 0.00 |
| Fugong County | 0.00 | 1.01 | 0.00 | 0.00 | 0.00 | 0.00 | 0.00 | 0.00 | 0.00 | 0.00 | 0.10 |
| Gongshan Derung and Nu Autonomous County | 0.00 | 0.00 | 0.00 | 0.00 | 0.00 | 0.00 | 0.00 | 2.58 | 0.00 | 0.00 | 0.26 |
| Lanping Bai and Pumi Autonomous County | 0.00 | 0.00 | 0.46 | 0.00 | 0.00 | 0.00 | 0.00 | 0.00 | 0.45 | 0.00 | 0.09 |
| **Dêqên Tibetan Autonomous Prefecture** | **1.99** | **0.74** | **0.25** | **0.49** | **0.97** | **0.98** | **1.68** | **1.21** | **1.21** | **0.97** | **1.05** |
| Shangri-la City | 0.57 | 0.00 | 0.00 | 0.57 | 0.57 | 0.57 | 0.56 | 0.56 | 0.55 | 0.56 | 0.45 |
| Dêqên County | 4.48 | 1.48 | 1.46 | 0.00 | 1.45 | 4.38 | 2.84 | 2.94 | 2.92 | 2.94 | 2.49 |
| Weixi Lisu Autonomous County | 2.47 | 1.24 | 0.00 | 0.61 | 1.21 | 0.00 | 2.39 | 1.22 | 1.21 | 0.61 | 1.10 |

*NCDR: new cases detected rate per 100,000 population.

(P = 0.00, 0.00, 0.00, and 0.065, respectively) are shown in Table 5 and Fig 4; there was no significant difference between the quaternary cluster and the other clusters (P = 0.0065) (Table 5).

The most likely cluster was mainly distributed in southeastern Yunnan Province and covered eleven counties (comprising Kaiyuan city, Mengzi city, Mile city, Pingbian Miao Autonomous County, Wenshan city, Yanshan County, Yunnan, West-chou, Malipo, Qiubei County, Guangnan County, Maguan County) and two districts (Honghe Hani and Yi Autonomous Prefecture, and Wenshan Zhuang and Miao Autonomous Prefecture), with a relative risk (RR) of 5.046515 and log likelihood ratio (LRR) of 271.749664 (P = 0.000). The cluster time was from January 1, 2011, to December 31, 2015 (Tables 5 and 6). In addition, one statistically significant secondary cluster and one statistically significant tertiary cluster, with high incidence rates of leprosy, were detected. The secondary and tertiary clusters were distributed in the southwestern and northern areas of Yunnan, respectively. The secondary cluster covered four counties (Menglian Dai, Lahu and Va Autonomous County; Lancang Lahu Autonomous County; Ximeng Va Autonomous County; and Menghai County) located in two districts (Pu'er city and Xishuangbanna Dai Autonomous Prefecture) (RR = 4.857894, LRR = 71.672149, and P = 0.000), and the tertiary cluster covered eleven counties (Luquan Yi and Miao Autonomous County, Yongsheng County, Huaping County, Mouding County, Yao'an County, Dayao County, Yongren County, Yuanmou County, Wuding County, Xiangyun County, and Binchuan County) in four districts (Kunming city, Lijiang city, Chuxiong Yi Autonomous Prefecture, and Dali Bai Autonomous Prefecture) (RR = 2.439682, LRR = 31.219025, and P = 0.000). The cluster times were from January 1, 2015, to December 31, 2019, and January 1, 2011, to December 31, 2014 (Tables 5 and 6).

## Discussion

In our study, we first performed a descriptive analysis of the epidemic situation of leprosy in Yunnan; then, we used spatial analysis methods to study spatial patterns and spatiotemporal clustering at the county level.

Both the number of newly detected leprosy cases and the NCDR in Yunnan decreased steadily during the ten-year study period. The NCDR of leprosy declined to 0.25 cases per 100,000 population in 2020 from 0.62 cases per 100,000 population in 2011. This downward trend was consistent with the results of studies in other provinces and cities [31] and the whole country [10]. This implies that the prevalence of leprosy in Yunnan has been controlled and remains at a low epidemic level. This achievement is due to the availability of MDT and

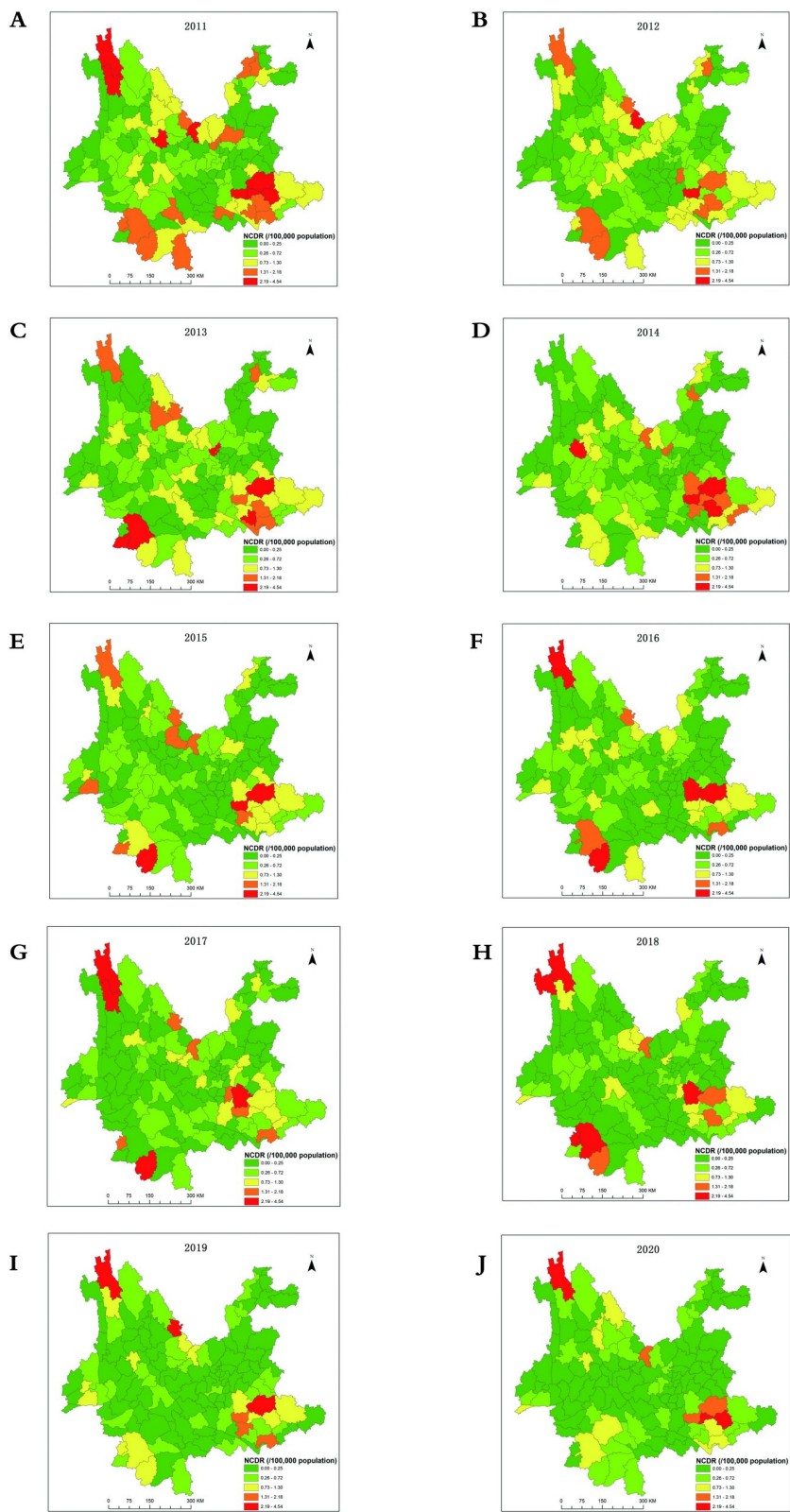

**Fig 2. Spatial Distributions of NCDRs of Leprosy Cases in Yunnan, China, 2011–2020. The base layer is from**
https://download.csdn.net/download/DEMservice/13272802?ops_request_misc=%257B%2522request%255Fid%
2522%253A%252216311694421678026 9845491%2522%252C%2522scm%2522%253A%252220140713.130102334.pc%

successful public health interventions. A special fund for leprosy was established in the region by the central government in 2004, and the leprosy elimination program (2011–2020) was funded by the Yunnan Province Government and the Health Administrative Department [11]. However, some regions with high NCDRs remain.

In this study, the majority of newly detected leprosy cases were classified as MB cases. According to some authors, the predominance of MB cases indicates late diagnosis and maintenance of the leprosy transmission chain [20]. Among the 1907 new cases diagnosed and geocoded in Yunnan, male patients predominated. This finding corroborates the findings of studies in China [10] and worldwide [32]. The proportion of females and children with leprosy fluctuated during the study period, suggesting underdiagnosis [33,34]. In addition, the proportion of physical disability in leprosy patients was more than 10%, which was very high and demonstrated that patients had advanced disease at the time of diagnosis, perpetuating the disease transmission chain [35].

According to the spatial distribution analysis, from 2011–2013, the distribution pattern of the NCDRs of leprosy showed significant spatial heterogeneity. In 2014, the high-NCDR regions were mainly located in southeastern, northern, and southwestern regions. After 2015, in addition to the above areas, the northwestern region became a new high-NCDR region. Annual spatial monitoring in endemic regions can significantly help to identify foci of leprosy and increase the degree and intensity of targeted health measures [20]. In this study, there were significant annual variations in the NCDRs among regions, which may significantly represent a lack of effective surveillance in low-epidemic areas and the presence of pseudosilent areas [2]. These abrupt changes in the epidemiological scenario of leprosy may reinforce the problem of local underdiagnoses and, in part, justify the results found in this study [36].

The global spatial autocorrelation results in this study indicated that leprosy in Yunnan had an obvious spatial cluster distribution. The local spatial autocorrelation analysis showed that the hot spots were located in the southeastern, northern and northwestern regions. The main hot spots of leprosy in southwestern Yunnan were basically consistent with the high-NCDR region of leprosy in Yunnan. The six districts, Qiubei County, Kaiyuan city, Mile city, Yanshan city, Wenshan city, and Mengzi city, located in the southeastern hot spot region, were the

**Table 3. The result of global spatial autocorrelation analysis of new detected leprosy cases in Yunnan, China 2011–2020.**

| Year | Moran's $I$ | z | p |
| --- | --- | --- | --- |
| 2011 | 0.203 | 4.299 | 0.000 |
| 2012 | 0.215 | 4.600 | 0.000 |
| 2013 | 0.218 | 4.675 | 0.000 |
| 2014 | 0.260 | 5.486 | 0.000 |
| 2015 | 0.186 | 4.056 | 0.000 |
| 2016 | 0.077 | 1.795 | 0.073 |
| 2017 | 0.098 | 2.176 | 0.030 |
| 2018 | 0.178 | 3.865 | 0.000 |
| 2019 | 0.204 | 4.394 | 0.000 |
| 2020 | 0.214 | 4.648 | 0.000 |

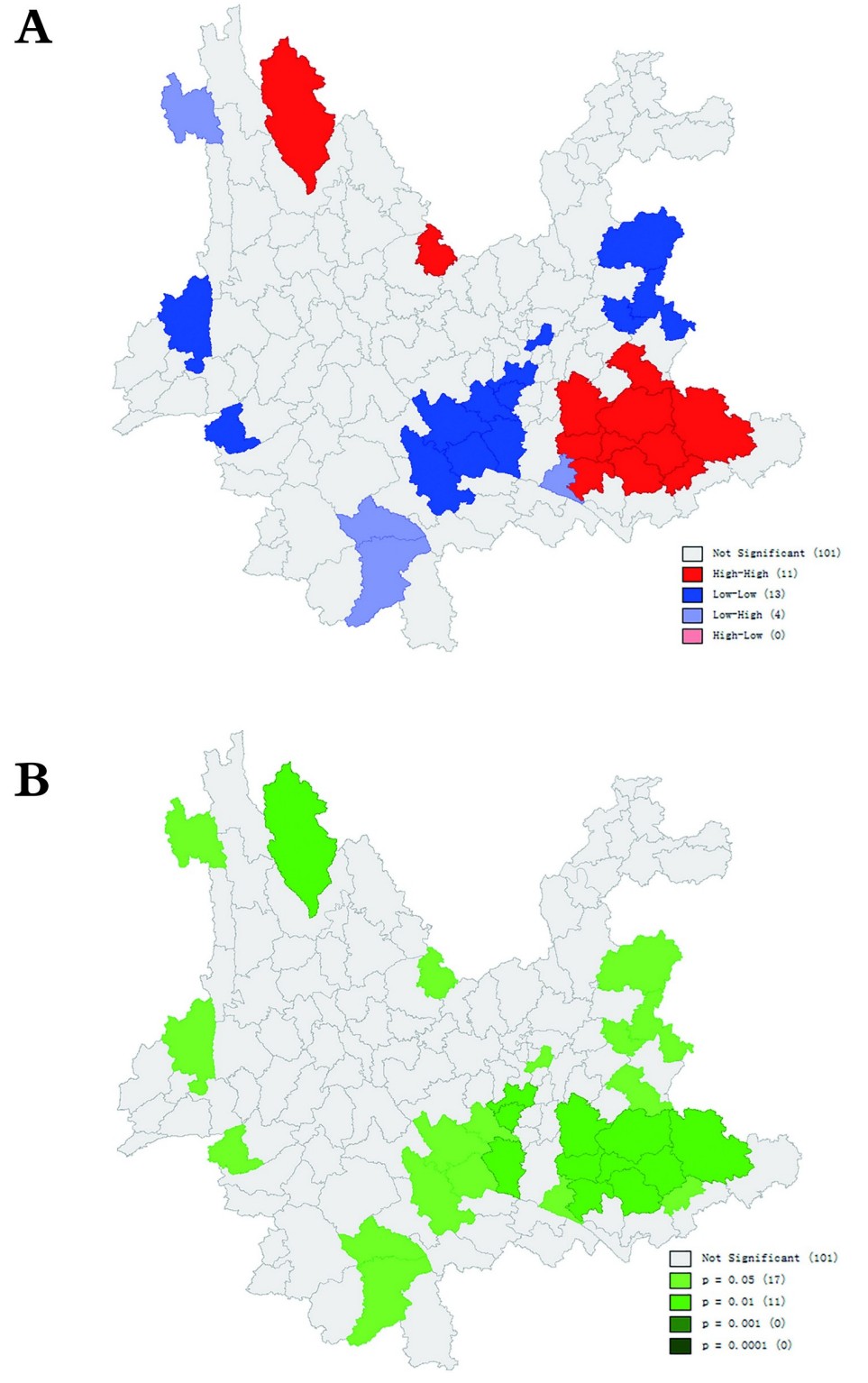

**Fig 3. Local Spatial Autocorrelation of Newly Detected Leprosy Cases in Yunnan, China, 2011–2020. The base layer is from** https://download.csdn.net/download/DEMservice/13272802?ops_request_misc=%257B%2522request%255Fid%2522%253A%2522163116944216780269845491%2522%252C%2522scm%2522%253A%252220140713.130102334.pc%255Fall.%2522%257D&request_id=163116944216780269845491&biz_id=1&utm_medium=distribute.pc_search_result.none-task-download-2~all~first_rank_ecpm_v1~rank_v29_ecpm-4-13272802.first_rank_v2_pc_rank_v29&utm_term=%E7%9C%81%E5%B8%82%E5%8E%BF%E5%8C%BA%E8%BE%B9%E7%95%8Cshp&spm=1018.2226.3001.4187.

**Table 4. The result of local spatial autocorrelation analysis of new detected leprosy cases in Yunnan, China 2011–2020.**

| Clusters Regions | Region | Prefecture | County |
|---|---|---|---|
| **High-high cluster** | Southeastern | Wenshan Zhuang and Miao Autonomous Prefecture | Wenshan City, Yanshan County, Xichou County, Qiubei County, Guangnan County, |
| | | Honghe Hani and Yi Autonomous Prefecture | Mengzi City, Kaiyuan City, Mile City |
| | | Qujing City | Shizong County, |
| | North | Chuxiong Yi Autonomous Prefecture | Yongren County |
| | Northeastern | Dêqên Tibetan Autonomous Prefecture | Shangri-La County |
| **Low-low cluster** | East | Baoshan City | Tengchong County |
| | Southeastern | Lincang City | Zhenkang County |
| | West | Qujing City | Xuanwei County, Fuyuan County, Qilin County |
| | Middle | Puer City | Mojiang Hani Autonomous County |
| | | Honghe Hani and Yi Autonomous Prefecture | Shiping County, |
| | | Yuxi City | Hongta county, Eshan Yi Autonomous County, Xinping Yi and Dai Autonomous County, Yuanjiang Hani, Yi and Dai Autonomous County, |
| | | Kunming City | Guandu county, Jinning County, |
| **Low-high cluster** | East | Honghe Hani and Yi Autonomous Prefecture | Gejiu City |
| | South | Pu'er City | Simao City, |
| | | Xishuangbanna Dai Autonomous Prefecture | Jinghong City |
| | Northeastern | Nujiang Lisu Autonomous Prefecture | Gongshan Derung and Nu Autonomous County |
| **High-low cluster** | Not detected | / | / |

areas with highest NCDRs of leprosy during the 10-year study period. Their annual average notification rates from 2011 to 2020 ranked first, third, sixth, seventh and eleventh, respectively.

The spatiotemporal scan analysis of leprosy cases from 2011 to 2020 showed that there were three clusters located in the southeastern, southwestern, and northern regions. The probably primary cluster was concentrated in southeastern Yunnan, covering two districts and eleven counties; the six counties described above were also involved. The cluster time period was from 2011 to 2015. This result implied that transmission was occurring in areas where minority ethnicities congregated and people had poor mobility.

This is the first analysis of the spatiotemporal cluster characteristics of newly detected leprosy cases at the county level in Yunnan, China. The results of the analysis identified areas at high risk of leprosy in the province. All spatial analysis techniques have advantages and disadvantages, and using complementary methods becomes necessary to achieve greater accuracy in analyzing priority areas for elimination [28].

**Table 5. The result of spatial-temporal analysis of new detected leprosy cases in Yunnan, China, 2011–2020.**

| Cluster Type | Number of Clustering areas | Observed cases | Expected cases | Relative risk | Log likelihood ratio | P value | Time frame |
|---|---|---|---|---|---|---|---|
| **Primary cluster** | 11 | 363 | 84.887 | 5.047 | 271.750 | 0.000 | 2011/01/01-2015/12/31 |
| **Secondary cluster** | 4 | 93 | 19.915 | 4.858 | 71.672 | 0.000 | 2015/01/01-2019/12/31 |
| **Tertiary cluster** | 11 | 107 | 45.360 | 2.440 | 31.219 | 0.000 | 2011/01/01-2014/12/31 |
| **Quaternary cluster** | 2 | 18 | 4.863 | 3.727 | 10.464 | 0.065 | 2011/01/01-2012/12/31 |

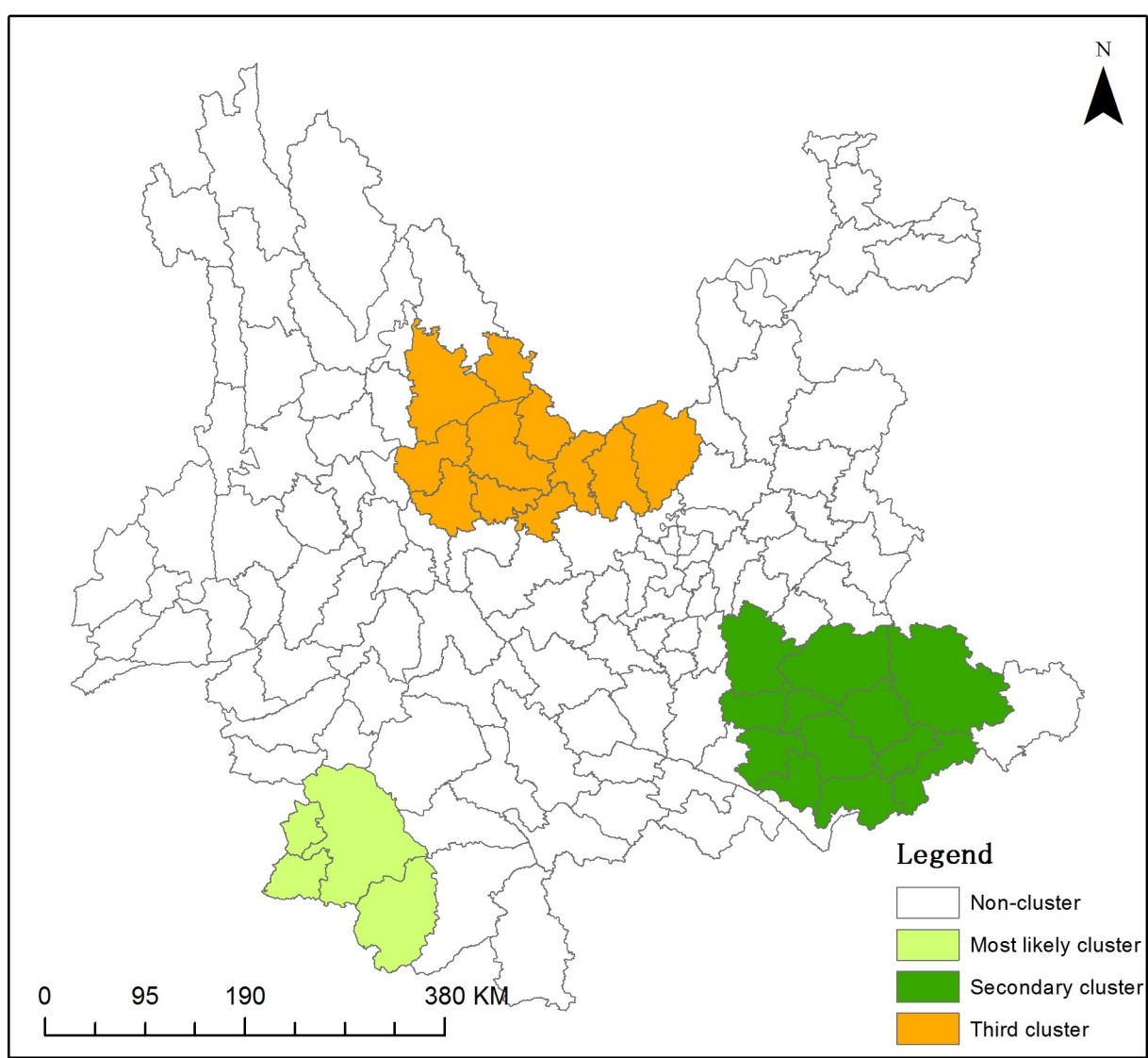

**Fig 4. Spatiotemporal Clusters of Newly Detected Leprosy Cases in Yunnan, China, 2011–2020. The base layer is from**
https://download.csdn.net/download/DEMservice/13272802?ops_request_misc=%257B%2522request%255Fid%2522%253A%
2522163116944216780269845491%2522%252C%2522scm%2522%253A%252220140713.130102334.pc%255Fall.%2522%257D&request_id=
163116944216780269845491&biz_id=1&utm_medium=distribute.pc_search_result.none-task-download-2~all~first_rank_ecpm_v1~rank_
v29_ecpm-4-13272802.first_rank_v2_pc_rank_v29&utm_term=%E7%9C%81%E5%B8%82%E5%8E%BF%E5%8C%BA%E8%BE%B9%E7%
95%8Cshp&spm=1018.2226.3001.4187.

This study has limitations. Potential risk factors, such as poverty [22,24], poor living conditions [2,21], inadequate access to medical services [2], household income [18], and coinfection with helminths [19], which has been previously reported to be associated with a high incidence of leprosy, were not evaluated in this study. More detailed analyses in the counties in the identified clusters should be analyzed to elucidate the disease profile and define more specific intervention targets and control strategies. The southeastern, northwestern, and northern regions of the spatial cluster bordered Guangxi Province, Tibet Autonomous Region, and Sichuan Province, respectively. Analysis of larger regions not limited to Yunnan Province could reveal the localized spatial and transmission characteristics of leprosy more comprehensively and accurately. In addition, analysis of statistics from 2020 would not reflect reality due to the

**Table 6. The identified significant high-rate spatial-temporal clusters of new detected leprosy cases in Yunnan, China, 2011–2020.**

| Most likely cluster | Secondary cluster | Thirdly cluster |
|---|---|---|
| **Honghe Hani and Yi Autonomous Prefecture** | **Pu'er City** | **Kunming City** |
| Kaiyuan City, | Menglian Dai, | Luquan Yi and Miao Autonomous County |
| Mengzi City, | Lahu and Va Autonomous County | **Lijiang City** |
| Mile City, | Ximeng Va Autonomous County | Yongsheng County |
| Pingbian Miao Autonomous County | **Xishuangbanna Dai Autonomous Prefecture** | Huaping County |
| **Wenshan Zhuang and Miao Autonomous Prefecture** | Menghai County | **Chuxiong Yi Autonomous Prefecture** |
| Wenshan City, | | Mouding County |
| Yanshan County, | | Yao'an County |
| Xichou County, | | Dayao County |
| Malipo county, | | Yongren County |
| Maguan County, | | Yuanmou County |
| Qiubei County, | | Chuxiong County |
| Guangnan County | | **Dali Bai Autonomous Prefecture** |
| | | Xiangyun County, |
| | | Binchuan County |

impact of the coronavirus disease 2019 (COVID-19) pandemic that occurred during this period [37].

## Conclusion

The overall number of cases of leprosy in Yunnan decreased; however, some regions maintained high NCDRs and/or clusters. Leprosy prevention and control efforts in Yunnan Province should be continuously implemented, and the prevention and control of leprosy in high-risk regions should be prioritized and further strengthened.

## Acknowledgments

We thank the leprosy clinicians at the Center for Disease Control and Prevention of Yunnan Province and the province of Yunnan, China, for their excellent work on the control and prevention of leprosy.

## Author Contributions

**Conceptualization:** Xiaohua Chen.

**Data curation:** Tie-Jun Shui.

**Formal analysis:** Xiaohua Chen.

**Funding acquisition:** Tie-Jun Shui.

**Investigation:** Xiaohua Chen.

**Methodology:** Xiaohua Chen.

**Project administration:** Tie-Jun Shui.

**Resources:** Tie-Jun Shui.

**Supervision:** Tie-Jun Shui.

**Validation:** Tie-Jun Shui.

**Visualization:** Xiaohua Chen.

**Writing – original draft:** Xiaohua Chen.

**Writing – review & editing:** Xiaohua Chen.

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
