## [Decision Letter · Decision Letter 0]

3 Aug 2021

Dear Mrs. tiejun Shui,

Thank you very much for submitting your manuscript "The state of the leprosy  epidemic in Yunnan , China 2011-2020: a spatial and spatial-temporal analysis and highlight areas for intervention" for consideration at PLOS Neglected Tropical Diseases. As with all papers reviewed by the journal, your manuscript was reviewed by members of the editorial board and by several independent reviewers. The reviewers appreciated the attention to an important topic. Based on the reviews, we are likely to accept this manuscript for publication, providing that you modify the manuscript according to the review recommendations. 

Sincerely,

Carlos Franco-Paredes

Associate Editor

Gerson Penna

Deputy Editor

Reviewer's Responses to Questions

**Key Review Criteria Required for Acceptance?**

**Methods**

-Are the objectives of the study clearly articulated with a clear testable hypothesis stated?

-Is the study design appropriate to address the stated objectives?

-Is the population clearly described and appropriate for the hypothesis being tested?

-Is the sample size sufficient to ensure adequate power to address the hypothesis being tested?

-Were correct statistical analysis used to support conclusions?

-Are there concerns about ethical or regulatory requirements being met?

Reviewer #1: Authors need to make adjustments to the statistical method.

Reviewer #2: 1. Although the study design is appropriate to address the stated objectives, however, how did the author consider some factors such as floating population of new cases which maybe cause the different results.

2. All the data from LEPMIS should be confirmed by the national and provincial experts.

Reviewer #3: Methods are appropriated to the objectives traced for the study.

**Results**

-Does the analysis presented match the analysis plan?

-Are the results clearly and completely presented?

-Are the figures (Tables, Images) of sufficient quality for clarity?

Reviewer #1: The results tables and figures need adjustments.

Reviewer #2: 1. Line 142-144"The number of newly detected leprosy cases were declined from 283 newly detected leprosy patients reported in 1990, to 119 cases reported at the end of 2020." Are the 283 newly detected leprosy patients reported in 2011?

2. All figures are not very clear, and should be more clear.

Reviewer #3: Yes for all questions

**Conclusions**

-Are the conclusions supported by the data presented?

-Are the limitations of analysis clearly described?

-Do the authors discuss how these data can be helpful to advance our understanding of the topic under study?

-Is public health relevance addressed?

Reviewer #1: Please, see "Summary and General Comments"

Reviewer #2: "The epidemic of leprosy in Yunnan decreased totally. While there were still part of regions with high NCDR and/or clustering regions." What's the reason? it didn't analyze very clear, especially in some areas with stable high NCDR.

Reviewer #3: Yes for all questions

**Editorial and Data Presentation Modifications?**

Reviewer #1: (No Response)

Reviewer #2: Minor Revision

Reviewer #3: (No Response)

**Summary and General Comments**

Reviewer #1: Abstract: 

According Author Guidelines: “The Abstract is conceptually divided into the following three sections with these headings: Background, Methodology/Principal Findings, and Conclusions/Significance.” 

Recommendations: Background is missing in manuscript.

Introduction:

According authors’ manuscript: 

(57-59) “Although the World Health Organization (WHO) elimination target has been achieved in 2000, 58 with a global prevalence rate of <1 case/10,000 people leprosy remains an important public health 59 problem in some low and middle-income countries” 

Consideration: Not all countries have achieved the goal of "eliminating leprosy" proposed by the WHO, including Brazil, which is a large country and has an expressive contribution to maintaining the number of leprosy cases. This reviewer suggests an introduction that discusses the aspects that led to the declaration of elimination of leprosy, especially in hyperendemic countries.

Statistical Analysis:

According authors’ manuscript: (121-122) “Grade 2 deformity was defined as visible disability”

Consideration: There is a conceptual inversion in this sentence. There is a conceptual inversion in this sentence. The correct sentence would be degree 2 of disability not as visible deformity.

The reference used was (30) WHO Expert Committee on Leprosy, World Health Organization. WHO Expert Committee on leprosy [meeting held in Geneva from 17 to 24 November 1987]: Sixth report. Geneva, 465 Switzerland: World Health Organization; 1988.

However, according WHO Expert Committee on Leprosy: seventh report (1).

“Committee endorsed this grading with the amendment that lagophthalmos. iridocyclitis and corneal opacities be considered as grade 2”.

This aspect is critical, was not taken into consideration, and may have led to underestimated results concerning the evaluated population. The same mistake is found in the caption of table 1.

Results and discussion:

According authors’ manuscript:

(151-154): “A total of 1279 MB cases were reported in Yunnan Province during 2011–2020, which accounted for 67.07% of new detected leprosy cases in the whole province. The number of MB cases was declined year by year from 172 in 2011 to 96 in 2020, while the ratio of MB cases was increasing year by year, which ranged from 60.78% in 2011 to 80.67% in 2020”

(160-161) “The rate of G2D gradually decreased to 10.34% in 2018, while increased to 12.50% in 2019, but further decreased to 10.08% in 2020”

(167-168) “The prevalence of leprosy among children fluctuated from 0.63% to 5.35%.”

(173-174) “The number of new cases of leprosy among female decreased from 103 in 2011 to 41 in 2020, with the proportion of female leprosy cases fluctuating between 29.95% and 38.51%”

(278-280) “According to spatial distribution analysis, from 2011-2013, the distribution pattern of the NCDR of leprosy showed significant spatial heterogeneity. In 2014, the high NCDR regions were mainly located in southeast, north, southwest regions.”

Consideration 1: The results found need better interpretation and discussion. The epidemiological-evolutionary course of leprosy is peculiar and does not change abruptly from one year to another, as shown in this manuscript. There are significant variations in the NCDR from one region to another year to year that may significantly represent a lack of effective surveillance in low-endemic areas and the occurrence of pseudo-silent areas. In the period between 2011 to 2020 there are areas with NCDR/100,000 inhabitants with averages between 2.19 and 4.54, becoming areas with NCDR/100,000 with averages between 0.00 and 0.25, and vice versa. (2).

Consideration 2: Grade 2 disability in the diagnosis of leprosy is another very sensitive epidemiological indicator. It amounted to around 10% between the years 2011 to 2020 in this region, demonstrating that patients remain with advanced disease at the time of diagnosis, perpetuating the disease transmission chain. (3)

Consideration 3: The woman leprosy diagnosis' in this region follows the same trend observed globally, however, the rates fluctuation' in the period was considerable, suggesting underdiagnoses. (4)

Consideration 3: Children with leprosy directly reflect the epidemiological scenario of the disease in a particular region. The large fluctuation in the period evaluated should be widely discussed, also suggesting underdiagnoses. (5).

The number of diagnosed children does not follow the proportion of new multibacillary cases diagnosed, according table one. What do the authors think about this?

(280-281): “After 2015, besides the above areas, northwest region became the new high NCDR region”.

Consideration 3: I would like to understand this result highlighted in the illustration provided in atached document.

Consideration 4: In regions with epidemiological clusters for leprosy, there is always an expectation of the new cases numbers diagnosed in a given year to the number of new cases that will be detection’ in the following year. These abrupt changes in the epidemiological scenario of leprosy may reinforce a problem of local underdiagnoses and, in part, justify the results found by the authors in this manuscript. (6)

Consideration 5: What does the number of “expected cases” expressed in table 5 mean?

Final considerations: 

The results of this manuscript are relevant and contribute to the proposed objective. The discussion needs to consider other aspects that promote improvement in local leprosy control actions. 

In this way, other locations where leprosy remains a public health problem can benefit these results. The number of leprosy newly cases remains practically stable over the last 12 years.

Obviously, the statistics for the years 2020 and 2021 will not reflect reality when available, taking into account the impact of the pandemic by COVID 19 and the lack of multidrug therapy that occurred during this period. 

This reviewer ends with a question, title of a manuscript referenced below: Global elimination of leprosy by 2020: are we on track? (7)

References:

1. WHO Technical Report Series No. 874. Geneva: WHO; 1998 (http://www.who.int/iris/handle/10665/42060).

2. Silva CLM, Fonseca SC, Kawa H, Palmer DOQ. Spatial distribution of leprosy in Brazil: a literature review. Rev Soc Bras Med Trop. 2017;50:439-449.

3. Raposo MT, Reis MC, Caminha AVQ, et al. Grade 2 disabilities in leprosy patients from Brazil: Need for follow-up after completion of multidrug therapy. PLoS Negl Trop Dis. 2018;12:e0006645.

4. Price VG. Factors preventing early case detection for women affected by leprosy: a review of the literature. Glob Health Action. 2017;10(sup2):1360550.

5. Oliveira MB, Diniz LM. Leprosy among children under 15 years of age: literature review. An Bras Dermatol. 2016;91:196-203.

6. Gupte MD, Murthy BN, Mahmood K, Meeralakshmi S, Nagaraju B, Prabhakaran R. Application of lot quality assurance sampling for leprosy elimination monitoring--examination of some critical factors. Int J Epidemiol. 2004;33:344-348.

7. Blok DJ, De Vlas SJ, Richardus JH. Global elimination of leprosy by 2020: are we on track? Parasit Vectors. 2015;8:548.

Reviewer #2: The study is very interesting, and will be benefit to the policy made of leprosy control in Yunnan province.

Reviewer #3: Dear Authors,

I read your paper with interest and attention. Knowledge about leprosy clustering is realy important in the manegement of resources and the adoption of adequate strategies for its control. I would like to point some questions to you:

In line 94 Yunnan is referred as a city, please verify

In your analysis and discussion you used the WHO classification, so I suggest to supress the information about RJ classification, mentioned in lines 104 and 105

In your results it could be interesting to inform how many children were diagnosed with G2D, if you have any in this period. It could be important to discuss this considering the new goal recomended by WHO. (lines 163- 169)

In line 313 southeast is repeated

In my opinion a brief discussion on G2D is lacking. The authors reported a decline in the number of NCDR with G2D, but this is still high. This finding is more robust than the ratio of MB case in the context of late diagnosis. I would like to hear from you about the rise of the ratio of MB cases in the context of progressive and rapid reducing of the disease in this province. In classical studies developed in Norway this finding was reported as a possible epidemiologial indicator of leprosy decline. So when we have a rising in the ratio of MB cases and a pronunced decline in NCDR with G2D maybe we are faced with a situation signaling to decline and not only with the situation of late diagnosis.

PLOS authors have the option to publish the peer review history of their article (what does this mean?). If published, this will include your full peer review and any attached files.

Reviewer #1: No

Reviewer #2: No

Reviewer #3: No

Figure Files:

Data Requirements:

Reproducibility:

References

---

## [Editor Report · Decision Letter 1]

1 Sep 2021

Dear Dr Shui;

We are pleased to inform you that your manuscript 'The state of the leprosy epidemic in Yunnan, China 2011–2020: a spatial and spatiotemporal analysis, highlighting areas for intervention' has been provisionally accepted for publication in PLOS Neglected Tropical Diseases.

Best regards,

Carlos Franco-Paredes

Associate Editor

Gerson Penna

Deputy Editor

---

## [Editor Report · Acceptance letter]

17 Sep 2021

Dear Mrs. Shui,

We are delighted to inform you that your manuscript, "The state of the leprosy epidemic in Yunnan, China 2011–2020: a spatial and spatiotemporal analysis, highlighting areas for intervention," has been formally accepted for publication in PLOS Neglected Tropical Diseases.

Best regards,

Shaden Kamhawi

co-Editor-in-Chief

Paul Brindley

co-Editor-in-Chief
